# Forestry Subsidies, Forestry Regulatory Policies, and Total Factor Productivity in Forestry—Plot-Scale Micro-Survey Data from A Heterogeneous Forest Types Perspective

**Lanfang Cao \*, Cheng Jiang, Qiqi Xiao, Tao Xu, Shuangshuang Lan, Jiali He and Shishi Peng**

School of Business, Central South University of Forestry and Technology, Changsha 410004, China
\* Correspondence: t20051248@csuft.edu.cn

**Abstract:** Enhancing the total factor productivity in forestry is an important part of deepening the reform of the collective forest rights system. Based on the survey data of 295 forest plots in 12 towns of Liuyang City, Hunan Province, China, the study utilized a three-stage DEA model to assess the total factor productivity of forestry at the plot level. The empirical study employs Tobit and fractional regression models to investigate the effects and differences of forestry subsidies and forestry regulatory policies on the heterogeneous total factor productivity of different types of forests. The study found that: (1) the mean value of plot-scale forestry total factor productivity is 0.127, and there are obvious differences in total factor productivity among timber forests, economic forests, and mixed forests; and (2) afforestation subsidies and nurturing subsidies significantly positively influence high-level TFP. Ecological benefit compensation positively affects high-level TFP, but is not significant at any level of TFP. Forestry regulatory policies negatively impact high-level TFP, but are not significant at any level of TFP. This paper puts forward countermeasure suggestions to improve forestry subsidy policies, optimize forestry regulatory policies, and improve forestry total factor productivity from the perspective of heterogeneous forest types.

**Keywords:** heterogeneous forest types; plot scale; total factor productivity

## 1. Introduction

The total factor productivity in forestry has garnered widespread attention worldwide. S. Kant (1997) studied the total factor productivity in the Canadian logging industry [1]. Daowei Zh. et al. (2006) compared and analyzed the trends in total factor productivity (TFP) in the sawmills and wood preservation industries (NAICS 3211) in the United States and Canada from 1958 to 2003 [2]. In recent years, an increasing number of scholars have studied forestry total factor productivity from either a macro-regional perspective or a micro-household perspective. However, the forestry production cycle is the fundamental time frame for calculating forestry total factor productivity, and it is only by using plots as the object of study that total factor productivity in forestry can be measured within a forest production cycle.

Under the current land property rights system in China, the ownership of forest land belongs to the state (or the public) or to the collective ownership of villages. Collective forests refer to forest land and trees owned collectively by villages, while collective forest areas refer to the areas where collective forests are located. Collective forests in China account for about 60% of the total forest area in the country, and collective forest areas play a very important role in China's ecological protection and economic development. The report of the 20th National Congress of the Communist Party of China explicitly proposed the deepening of the reform of the collective forest rights system. In 2023, the General Office of the Communist Party of China and the State Council issued the "Implementation Plan for Deepening the Reform of Collective Forest Rights System", which emphasizes improving the total factor productivity of forestry as an important content of deepening the reform of

the collective forest rights system. After the new round of collective forest rights reform from 2008 to 2012, household contracting became the main form of forestry management in collective forest regions [3]. However, it is relatively common for households to manage multiple non-adjacent small-area plots. The opportunity cost of labor migration further suppresses households' enthusiasm for investing in forestry production, resulting in low total factor productivity in collective forest areas [4]. Although the government initiated the "tripartite entitlement system" reform to encourage the transfer of collective forest production rights in an attempt to promote the appropriate scale of operation to improve the efficiency of forestry production, the general willingness of farmers to transfer is not high, and there are farmers with multiple pieces of forest land, small-scale operations are still commonplace [5]. Therefore, on the premise of strictly guaranteeing ecological security, the state stimulates the enthusiasm of households for forest management through a series of forestry subsidy policies, thereby improving the total factor productivity of forestry.

At present, the contradiction between forest resource protection and utilization in collective forest regions remains prominent. Forestry subsidies and forestry regulatory policies in parallel are prone to overlapping, resonating, or offsetting effects and constraints. On the one hand, forestry subsidies can reduce production costs and incentivize farmers to optimize factor allocation, leading to an increased total factor productivity [6]. On the other hand, strict forestry regulatory policies can increase transaction costs and lower farmers' income expectations, potentially reducing the scale of inputs such as labor and capital [7,8]. For instance, studies by Yang et al. (2018) demonstrate that afforestation subsidies have a promoting effect on farmers' investment in planting and nurturing stages, while the impact of forestry regulatory policies is not significant [9]. However, the input–output characteristics of economic forests, timber forests, and other forest types are very different, and there is a paucity of research on the effects of forestry subsidies and forestry regulatory policies on total factor productivity and differences in heterogeneous forest types.

In summary, existing studies have explained the relationship between forestry subsidies, forestry regulatory policies, and total factor productivity in forestry, but there is still some room for expansion: (1) With regard to the scale of the study, the plot is the basic unit of forestry production and operation, subsidy distribution, and forestry regulatory policies, and there are significant differences in the area, fertility conditions, subsidy amount, and regulatory restrictions of different plots operated by the same farmer [10]. However, existing studies treat "multiple plots" as "one piece of land", and the accuracy of the results of total factor productivity in forestry at the household scale needs to be further improved. (2) In terms of research data, studies related to total factor productivity in forestry at the farm household scale are generally cross-sectional data or short panel data [6], which are inconsistent with the natural attributes of a long forestry production cycle, which may lead to the deviation of research results from the actual situation. To scientifically measure total factor productivity in forestry, it is necessary to have input–output data for a complete production cycle. (3) In terms of research perspectives, the production characteristics of timber forests, economic forests, and other forest types are fundamentally different. Existing studies treat "heterogeneous forest types" as "homogeneous forest types" [11], which may lead to insufficient accuracy in measuring the total factor productivity of forestry and studying the effects of forestry policies. (4) In terms of research content, the existing literature mainly focuses on the impact of either subsidies or forestry regulatory policies, with insufficient attention paid to the combined effects of both policies. In view of this, this paper, from the perspective of forest type heterogeneity, takes the plot as the research scale, measures the total factor productivity of forestry in a complete production cycle at the plot scale, and reveals the impacts of forestry subsidies and forestry regulatory policies on the total factor productivity of timber forests, economic forests, and mixed forests, and the differences between them, so as to provide theoretical references for deepening the reform of the collective forest rights system and promoting the high-quality development of forestry. In this article, heterogeneous forest types refer to timber forests, economic forests, and mixed forests. Additionally, it does not involve biodiversity. Please refer to Table 1.

**Table 1.** The properties of the three forest types in this study.

| Forest Types | The Properties of Forest Types |
|---|---|
| Timber forest | Forests primarily cultivated for timber or bamboo, including species such as pine and fir et al. |
| Economic forest | Forests whose primary purpose is the production of fruits, edible oilseeds, industrial raw materials, and medicinal herbs. Economic forests are of high value and are generally intensively managed, including species such as camellia oleifera et al. |
| Mixed forest | Mixed forests include both economic forests and timber forests. |

## 2. Theoretical Analysis and Research Hypotheses

The total factor productivity is the output efficiency of all observable inputs within the production cycle, and it is an important efficiency indicator extending from manufacturing to other industries. Academia has introduced it into the field of agricultural and forestry economics, using the total factor productivity to reflect regional development quality and household production efficiency. The existing literature on the total factor productivity of forestry mainly includes the following: First, the measurement of the total factor productivity in forestry is as follows: existing studies use methods such as data envelopment analysis (DEA) or stochastic frontier analysis (SFA) to measure the total factor productivity at the macro-regional scale. Dandan G. et al. (2021) applied the three-stage data envelopment analysis (DEA) model to calculate the total factor productivity of 31 provinces in China from 2009 to 2018. Their study found that the total factor productivity of China's forestry sector has been continuously increasing [12]. Chen Ch. et al. (2023) found that, from 2013 to 2019, the annual average growth rate of China's forestry green total factor productivity was 5.03%, maintaining an overall stable growth trend [13]. The second is the study of factors affecting total factor productivity in forestry, which mainly includes forestry property rights systems and policies, natural factors, and the digital economy [14,15]; Mingming J. et al. (2023) found the integration of China's forestry industry can improve total factor productivity [16]. Zhong, S. et al. (2021) found that the impact of forestry total factor productivity on $CO_2$ emissions follows an "inverted U-shaped" curve, with a turning point at 0.9395 [17]. Thirdly, in order to enhance the total factor productivity of forestry, academics recommend that the promotion of an appropriate scale operation, technology Internet and platform Internet, developing financial services, and upgrading the quality of labor can improve the total factor productivity of forestry [18,19]. However, the input–output characteristics of forest types such as timber and economic forests are significantly different, making it difficult to accurately measure the total factor productivity of different forest types at either the macro-regional or micro-farmer scales. Therefore, this study measures the total factor productivity of forests at the plot scale and proposes the following research hypotheses:

**H1.** *There may be differences in the total factor productivity of forest plots under the heterogeneous forest types perspective.*

Forestry subsidies are a general term for public finance support measures led by the government to assist forestry operators [20]. The "Measures for the Management of Central Government Financial Forestry Subsidies" implemented in 2014 clarified that forestry subsidies are expenses for afforestation, nurturing, and ecological benefit compensation. The impact of forestry subsidies on total factor productivity in forestry mainly includes the following: first, subsidy funds are directly transferred to households based on plot size and dominant functions, which reduces the cost of production and operation per unit area of forest land; and, second, the subsidy thresholds and acceptance standards set higher requirements for farmers' business behaviour, which helps to improve the efficiency of forestry production [4,21]. However, timber forests, economic forests, and other forest types have very different input factors, input cycles, and other characteristics—do different subsidies make a difference to the total factor productivity of heterogeneous forest types?

This paper focuses on the impact of three types of forestry subsidies, namely, afforestation subsidies, nurturing subsidies, and ecological benefit compensation, on the total factor productivity of timber forest, economic forest, and mixed forest, and puts forward the following hypothesis:

**H2.** *Forestry subsidies have a positive impact on total factor productivity in forestry, and the impact may vary among heterogeneous forest types.*

Forestry regulatory policies are government restrictions on forest resource utilization, and the core of forestry regulatory policies is the quota management of timber harvesting. Numerous studies, through theoretical interpretations and empirical research, show that forest regulatory policies are an important factor affecting total factor productivity in forestry [7,22]. However, there are significant differences in the impact of forestry regulatory policies on heterogeneous forest types. Based on this, the following research hypothesis is proposed:

**H3.** *Forestry regulatory policies have a negative impact on total factor productivity in forestry, but the impact may vary among heterogeneous forest types.*

## 3. Research Design

### 3.1. Theoretical Model

This paper measures the total factor productivity (TFP) of forestry based on input–output data of plots in one production cycle. It then investigates the effects and differences of subsidies and forestry regulatory policies on the total factor productivity of timber forests, economic forests, and mixed forests. In China, forests are classified as commercial or public welfare based on their functional attributes. Public welfare forests prioritize ecological benefits. Commercial forests prioritize economic benefits and include timber forests, economic forests, and other forest types. Timber forests and economic forests are general terms for related tree types, and mixed forests include both economic and timber forests. Please refer to Table 1.

We establish the theoretical model as follows:

$$\text{LN (TPF + 1)} = F \text{ (forestry subsidies, forestry regulatory policies, control variables)} + \text{random disturbance term} \quad (1)$$

$$\text{TPF} = Y \text{ (total forestry output)} - F \text{ (land input, capital input, labor input)} - \text{environmental factors and random error term} \quad (2)$$

The dependent variable in Equation (1) is the plot-scale total factor productivity in forestry measured according to Equation (2), and the measured value of plot-scale total factor productivity in forestry is between 0 and 1. Taking the logarithm can reduce the extreme variance of the dependent variable, but it takes a negative value, so this paper adds 1 to the plot-scale total factor productivity, and then takes the logarithm. The key explanatory variables include two categories of forestry subsidies and forestry regulatory policies, of which forestry subsidies include afforestation subsidies, nurturing subsidies, and ecological benefit compensation, and forestry regulatory policies mainly refer to the forest harvesting quota management system. The control variables include plot fertility conditions, plot infrastructure, plot transportation conditions, whether or not to participate in forest insurance, household income level, and forest type.

### 3.2. Definition of Variables

(1)    Dependent variables

This paper establishes a theoretical model based on the Cobb–Douglas production function theory (2), selects variable indicators from input, output, and environment, and then uses the three-stage DEA model to calculate the total factor productivity in forestry, and the definition and descriptive analysis of each indicator are shown in Table 2.

**Table 2.** Definition and descriptive analysis of indicators for measuring total factor productivity in the forestry industry.

| Variable Type | Variable Name | Variable Definition | Average Value | Standard Deviation | Minimum Value | Maximum Value |
|---|---|---|---|---|---|---|
| Input variable | land input | Parcel size (in acres) | 28.96 | 49.36 | 0.2 | 320 |
| | labor input | Full-cycle labor input for plots (in man-days) | 603.10 | 2626.45 | 7 | 32,854 |
| | capital investment | Costs of full-cycle fertilizers, seedlings, etc., for plots (unit: yuan) | 69,456.86 | 278,240.10 | 50 | 4,124,000 |
| Output variable | total forest land output | Full-cycle output value of parcels (unit: yuan) | 104,661.01 | 163,729.58 | 1020 | 990,000 |
| Environment variable | natural disaster | Number of full-cycle disaster occurrences on plots | 2.90 | 0.75 | 1020 | 4 |
| | fertility conditions | Plot fertility conditions (1 = poor; 2 = fair; 3 = better; 4 = good) | 2.9220 | 0.7405 | 1 | 4 |
| | infrastructure | Whether the development of infrastructure such as roads, electricity, water, and networks in forest areas can meet the needs (1 = no; 2 = partially; 3 = yes) | 2.56 | 0.46 | 1 | 3 |
| | transportation condition | Accessibility of parcels (1 = not accessible; 2 = accessible) | 1.6339 | 0.4826 | 1.25 | 2 |
| | economic condition | Level of household income in the village (1 = low; 2 = medium; 3 = high) | 1.87 | 0.53 | 1 | 3 |

Note: Considering the long time span of the full cycle of the plots, this paper uses the year of the survey as the base period for the input–output variables, removing the effect of the time value of the data.

(2)  Core independent variables

The core independent variables include forestry subsidies (including afforestation subsidies, nurturing subsidies, and ecological benefit compensation) and forestry regulatory policies. In this paper, whether or not a plot receives an afforestation subsidy, whether or not a plot receives a conservation subsidy, and whether or not a plot receives ecological benefit compensation are used as proxy variables for forestry subsidy. Whether a plot applies to the cutting index as a proxy variable for forestry regulatory policies, the definition and descriptive analysis of each index are shown in Table 3.

**Table 3.** Definition and descriptive analysis of explanatory variables.

| Variable Name | Variable Definition | Sample Size (Statistics) | Average Value | Standardized Value | Minimum Value | Maximum Value |
|---|---|---|---|---|---|---|
| Afforestation subsidies | Whether the plot receives afforestation subsidies (0 = no; 1 = yes) | 295 | 0.1763 | 0.3817 | 0 | 1 |
| Nurturing subsidies | Whether the plot receives a conservation subsidy (0 = no; 1 = yes) | 295 | 0.1661 | 0.3728 | 0 | 1 |
| Ecological benefit compensation | Whether the parcel receives ecological benefit compensation (0 = no; 1 = yes) | 295 | 0.0678 | 0.2518 | 0 | 1 |
| Forestry regulatory policies | Whether the plot is applying for a harvesting target (0 = no; 1 = yes) | 295 | 0.4441 | 0.4977 | 0 | 1 |

### 3.3. Methods of Estimation

The study's data analysis comprises two parts. The first part utilizes a three-stage DEA model to calculate TFP. The second part of the study employs Tobit regression to analyze the factors that influence TFP, with a focus on forestry subsidies and forestry regulatory policies.

In order to accurately measure the total factor productivity of forestry at the plot scale, according to model (2), this paper adopts the three-stage DEA model proposed by Fried et al. [23]. The model can make up for the traditional DEA model and does not take into account the environmental factors. Random noise on the efficiency evaluation of the defects of the measured total factor productivity can more truly reflect the internal management level of the decision-making unit.

Fried (1999, 2002) noted that traditional DEA models do not account for the impact of environmental factors and random noise on the evaluation of decision-making unit efficiency. In his two subsequent articles, he discusses how to incorporate environmental factors and stochastic noise into DEA models [23,24]. Among them, the first paper only eliminated environmental factors, while the latter considered both eliminating environmental factors and random noise, which is known as the three-stage DEA model in academia. The first and third stages of the "three-stage DEA model" are no different from the traditional DEA model; the key lies in how to eliminate environmental factors and random noise in the second stage. This is carried out as follows:

Step 1: Calculate the total factor productivity (TFP) value including environmental factors and random noise by using input variables (such as land, capital, labor, etc.) and output variables.

Step 2: Use the stochastic frontier analysis (SFA) regression function to eliminate environmental factors and random noise from input variables (such as land, capital, labor, etc.), obtaining new values for input variables. According to the ideas of Fried et al., we can regress environmental variables and mixed error terms using slack variables [23,24]. First, construct a similar SFA regression function (taking input orientation as an example): $S_{ni} = f(Z_i; \beta_n) + v_{ni} + \mu_{ni}; i = 1, 2, \ldots, I; n = 1, 2, \ldots, N$, where $S_{ni}$ is the slack value of the Nth input of the $i$ decision-making unit; $Z_i$ and $\beta_n$, respectively, represent the value and coefficient of the environmental variable; $v_{ni} + \mu_{ni}$ is the mixed error term; $v_{ni}$ indicates random disturbance, and $\mu_{ni}$ indicates managerial inefficiency. Next, we follow the approach of Jondrow et al. (1982), Luo Dengyue (2012), and Chen Weiwei et al. (2014) [25–27], based on the separation formula $E(\mu|\varepsilon) = \sigma_* \left[ \frac{\phi\left(\lambda\frac{\varepsilon}{\sigma}\right)}{\Phi\left(\frac{\lambda\varepsilon}{\sigma}\right)} + \frac{\lambda\varepsilon}{\sigma} \right]$, to calculate managerial inefficiency $\mu_{ni}$, and, finally, based on the formula $X_{ni}^A = X_{ni} + \left[ max\left(f\left(Z_i; \overset{\wedge}{\beta}_n\right)\right) - f\left(Z_i; \overset{\wedge}{\beta}_n\right)\right] + [max(v_{ni}) - v_{ni}]; i = 1, 2, \ldots, I; n = 1, 2, \ldots, N$. The adjusted values of the original inputs and relaxation variables are summed to obtain the adjusted input values $X_{ni}^A$.

Step 3: Utilize the input variable values $X_{ni}^A$ obtained from the second stage, and, once again, employ DEA to calculate the total factor productivity (TFP) value without considering environmental factors and random noise.

### 3.4. Data Sources

Liuyang City, Hunan Province is a national collective forest rights system reform demonstration area in China, which is typical and representative of the study. The research data of this paper come from the field survey data in July 2022 in 12 townships of Liuyang City, Hunan Province, namely, Guanqiao, Pu trace, Zhangfang, Gugang, Dahu, Dawishan, Yonghe, Guandu, Yanxi, Gaoping, Gongcheng, and Zhentou. The questionnaires targeted farmers' plots and collected data on the production characteristics of timber forests, economic forests, mixed forests, and other forest types. The production cycle of timber forests is divided into the afforestation stage, middle and young forests nurturing stage, management stage, and harvesting stage. The production cycle of economic forests is divided into the afforestation and nurturing stage, early fruiting stage, full fruiting stage, and declining

stage. For this study, sample farmers were selected using random sampling. A total of 252 valid farmer questionnaires were collected. After excluding recently added sample farmers and unused or abandoned forest land, 295 pieces of forest land were obtained. Of these, 107 were timber forests, 152 were economic forests, and 36 were mixed forests (refer to Table 4).

**Table 4.** Descriptive analysis of the sample.

| | Plot Definition | Number of Plots | Mean | Std.Dev. | Min | Max |
|---|---|---|---|---|---|---|
| Timber forest plots | Plots planted mainly with timber forests such as fir and pine | 107 | 32.24 | 50.14 | 1 | 240 |
| Economic forest plots | Plots mainly planted with economic forests such as oil tea and bamboo | 153 | 27.70 | 52.59 | 0.2 | 320 |
| Mixed forest plots | Plots with mixed fir, pine, bamboo, and shrub forests | 36 | 24.50 | 30.48 | 2 | 120 |

Note: The mean (Mean), minimum (Min), and maximum (Max) values in this table are in acres.

## 4. Descriptive Statistical Analysis

### 4.1. Descriptive Analysis of Forestry Subsidies, and Forestry Regulatory Policies

As for forestry subsidies, 52 plots received afforestation subsidies, accounting for 17.63% of the total plots; 49 plots received nurturing subsidies, accounting for 16.61% of the total plots; and 20 plots received ecological benefit compensation, accounting for 6.78% of the total plots. According to Table 5, in terms of afforestation subsidies, the number of plots of economic forests receiving afforestation subsidies is more than twice that of timber forests: 36 plots of economic forests received afforestation subsidies, accounting for 69.23%; 15 plots of timber forests received afforestation subsidies, accounting for 28.85%; and only 1 plot of mixed forests received afforestation subsidies. From the viewpoint of nurturing subsidies, the proportion of plots of timber forest and economic forest receiving nurturing subsidies is the same. For nurturing, 24 plots of economic forests (48.98%), 22 plots of timber forests (44.90%), and 3 plots of mixed forests (6.12%) were subsidized. Regarding ecological benefit compensation, 9 plots of timber forests and 10 plots of mixed forests received compensation, while only 1 plot of economic forest received compensation. It is worth stating that, according to the Forest Law of the People's Republic of China, economic forests, timber forests, and energy forests belong to commercial forests, and protection forests and special-use forests belong to public welfare forests; however, public welfare forests are constantly being adjusted in practice according to the actual situation. Therefore, the three types of forest types mainly discussed in this paper all have plots to receive ecological benefit compensation; i.e., there are public welfare forest plots for timber forests, economic forests, and mixed forest types.

Regarding forestry regulatory policies, this study used whether the plots had applied for logging targets as a proxy variable for forestry regulatory policies: 131 plots had applied for logging targets, accounting for 44.41% of the total plots. According to Table 5, the proportion of timber forest plots and economic forest plots that had applied for logging targets was 43.51% and 48.09%, respectively. According to the survey, the reasons why forest harvesting applications also exist in the economic forest plots in this study are as follows: firstly, economic forest plots also need to apply for harvesting indicators when renewing economic forests or adjusting economic forest types; and, secondly, bamboo forests are, by default, considered as a kind of economic forest in practice, and logging needs to comply with the technical regulations for forest harvesting (Article 56 of the 2019 Revised Forest Law [28]).

**Table 5.** Descriptive analysis of forestry subsidies, and forestry regulatory policies.

| | Afforestation Subsidies | | Nurturing Subsidies | | Ecological Benefit Compensation | | Forestry Regulatory Policies | |
|---|---|---|---|---|---|---|---|---|
| | Number of plots | Percentage | Number of plots | Percentage | Number of plots | Percentage | Number of plots | Percentage |
| Timber forest plots | 15 | 28.85% | 22 | 44.90% | 9 | 45.00% | 57 | 43.51% |
| Economic forest plots | 36 | 69.23% | 24 | 48.98% | 1 | 5.00% | 63 | 48.09% |
| Mixed forest plots | 1 | 1.92% | 3 | 6.12% | 10 | 50.00% | 11 | 8.40% |
| Total | 52 | 100% | 49 | 100% | 20 | 100% | 131 | 100% |

Note: Percentage in this table refers to the proportion of plots receiving a particular type of subsidy to the total number of plots receiving that type of subsidy.

*4.2. Descriptive Analysis of Total Factor Productivity in Forestry*

This article uses a three-stage DEA model to calculate the total factor productivity of forest plots at the plot level after excluding the influence of environmental factors. Based on the measurement results, the total factor productivity of forestry at the total sample plot scale decreased from 0.184 in the first stage to 0.127 in the third stage. This suggests that environmental factors have a significant impact on the total factor productivity of forestry. A comparison with the results in the established literature found that the relationship between total factor productivity in forestry at different research scales is as follows—provincial scale > farmer scale > plot scale—and the total factor productivity in forestry at the plot scale measured in this paper is the lowest.

Table 6 shows that the total factor productivity of heterogeneous forest types at the plot scale is ranked as follows: timber forest > economic forest > mixed forest. This significant difference confirms Hypothesis 1. Possible reasons for the differences in plot-scale total factor productivity among forest types may be that timber forests require specific factor inputs only during the afforestation and nursery stages of young and medium-sized forests. Additionally, they require minimal care when they enter the natural growth stage. Economic forests require intensive capital and labor inputs throughout the production cycle, and the output of small-scale operations is limited. The intensive management level of mixed forests may be lower than that of timber forests and economic forests.

**Table 6.** Descriptive analysis of total factor productivity in forestry.

| | Mean Value | Minimum Value | Maximum Value |
|---|---|---|---|
| Total sample plots | 0.1270 | 0.002 | 1 |
| Timber forest plots | 0.1711 | 0.003 | 1 |
| Economic forest plots | 0.1112 | 0.002 | 1 |
| Mixed forest plots | 0.0671 | 0.003 | 0.458 |

**5. Analysis of Empirical Results**

The study first conducts Tobit regressions on the total sample, timber forest sample, economic forest sample, and mixed forest sample to explore whether there are differences in the effects of forestry subsidies and regulatory policies on different types of forest plots. As forestry total factor productivity (TFP) values fall within the range of 0 to 1, exhibiting a bi-directional truncation feature, the Tobit model is suitable for analysis. The estimation results refer to Table 7.

According to the total sample model (1) in Table 7, for forestry subsidies, afforestation subsidies are significantly and positively related to plot-scale total factor productivity at the 1% confidence level. Nurturing subsidies are significantly positively correlated with

the dependent variable at the 1% confidence level. Ecological benefit compensation is positively and insignificantly related to the dependent variable. For forestry regulatory policies, forestry regulatory policies were negatively and non-significantly related to plot-scale total factor productivity. The empirical results illustrate that forestry subsidies can reduce the production and operation costs of plots, motivate farmers to expand forestry factor inputs, and have a significant role in promoting total factor productivity in forestry. Forestry regulatory policies have a negative effect on total factor productivity in forestry, but it is not statistically significant.

**Table 7.** Tobit regression results.

| | (1) Total Sample | (2) Timber Forest Sample | (3) Economic Forest Sample | (4) Mixed Forest Sample |
|---|---|---|---|---|
| Core explanatory variables | | | | |
| Afforestation subsidies | 0.0621 *** | 0.0806 * | 0.0525 ** | 0.315 *** |
| | (3.09) | (1.84) | (2.24) | (4.74) |
| Nurturing subsidies | 0.0554 *** | 0.0499 | 0.0542 * | 0.0648 |
| | (2.67) | (1.30) | (1.91) | (1.46) |
| Ecological benefit compensation | 0.0214 | −0.0109 | −0.0201 | 0.00847 |
| | (0.71) | (−0.20) | (−0.17) | (0.31) |
| Forestry regulatory policies | −0.00712 | −0.0245 | 0.00246 | 0.0293 |
| | (−0.46) | (−0.82) | (0.12) | (1.20) |
| Control variable | | | | |
| Plot geotechnical conditions | −0.0194 * | −0.0269 | −0.0281 * | 0.0305 * |
| | (−1.71) | (−1.15) | (−1.86) | (1.97) |
| Parcel infrastructure | 0.00557 | 0.0204 | 0.0102 | −0.0335 |
| | (0.32) | (0.62) | (0.43) | (−1.41) |
| Plot transportation conditions | 0.00610 | −0.0179 | 0.0122 | 0.0926 *** |
| | (0.36) | (−0.58) | (0.53) | (3.22) |
| Household income level | 0.0551 *** | 0.0758 *** | 0.0512 *** | 0.0319 |
| | (3.80) | (2.74) | (2.66) | (1.34) |
| Whether the parcel is insured or not | −0.0358 | −0.00990 | −0.0827 * | 0.0219 |
| | (−1.25) | (−0.20) | (−1.84) | (0.60) |
| Forest typies | −0.0450 *** | -- | -- | -- |
| | (−3.79) | | | |
| Intercept term | 0.0916 | 0.0132 | 0.00391 | −0.0734 |
| | (1.30) | (0.10) | (0.04) | (−0.77) |
| Sigma_e | 0.0160 *** | 0.0205 *** | 0.0142 *** | 0.00377 *** |
| | (12.14) | (7.31) | (8.72) | (4.24) |
| LR chi2(10) | 55.90 | 17.35 | 30.31 | 28.31 |
| Prob > chi2 | 0.000 | 0.043 | 0.000 | 0.000 |
| N | 295 | 107 | 152 | 36 |

Note: t statistics in parentheses, * $p < 0.1$, ** $p < 0.05$, *** $p < 0.01$.

According to model (2) of the timber forest sample in Table 7, in terms of forestry subsidies, afforestation subsidies are significantly and positively associated with the total factor productivity of timber forest plots at the 10% confidence level. Nurturing subsidies are positively and insignificantly related to the dependent variable. The dependent variable did not show a significant negative relationship with ecological benefit compensation. This may be due to the fact that the ecological benefit compensation was given to public welfare forest plots, which are subject to strict logging controls. This limits the incentives

for farmers to produce timber forest plots. In relation to forestry regulatory policies, the correlation between forestry regulatory policies and the total factor productivity of timber forests is negative and insignificant. This suggests that forestry regulatory policies have a negative impact on the total factor productivity of timber forests, but the impact is not statistically significant.

According to model (3) of the economic forest sample in Table 7, in terms of forestry subsidies, afforestation subsidies are significantly and positively related to total factor productivity in economic forests at the 5% confidence level. Nurturing subsidies are significantly positively correlated with the dependent variable at the 10% confidence level. The negative and insignificant correlation of ecological benefit compensation on the dependent variable indicates that ecological benefit compensation has a negative effect on the all-important productivity of economic forests. Regarding forestry regulatory policies, the study found that forestry regulatory policies have a positive but non-significant relationship with the total factor productivity of economic forests. This suggests that forestry regulatory policies do not have a negative impact on the total factor productivity of economic forests. This may be due to the fact that logging is not the primary source of income for economic forest plots, such as camellia oleifera. Additionally, the system of licenses for bamboo logging has been abolished, except for nature reserves. In 2014, the Opinions of the State Forestry Administration on Further Reform and Improvement of Collective Forest Harvesting Management (Lin Zifa (2014) No. 61 [29]) proposed that forest logging permits should not be issued for bamboo harvesting for the time being, and, in 2019, the newly amended Forestry Law explicitly proposed to abolish the bamboo logging permit system (except for nature reserves).

According to the mixed forest model (4) in Table 7, for forestry subsidies, afforestation subsidies are significantly and positively correlated to total factor productivity in mixed forests at the 1% confidence level. Nurturing subsidies are positively and insignificantly related to the dependent variable. Ecological benefit compensation is positively and insignificantly related to the dependent variable. Regarding forestry regulatory policies, forestry regulatory policies are positively and insignificantly related to total factor productivity in mixed forests.

In summary, in terms of forestry subsidies, afforestation subsidies and nurturing subsidies have a positive impact on the total factor productivity of all types of forest types to varying degrees of significance, while ecological benefit compensation has a negative effect on the total factor productivity of timber forests and economic forests, but a positive effect on the total factor productivity of mixed forests. This supports research Hypothesis 2. Forestry regulatory policies affect the total factor productivity of timber forests in the opposite direction to the total factor productivity of economic and mixed forests, testing research Hypothesis 3.

To further investigate the direct effects of forestry subsidies and regulatory policies on different levels of TFP, this study employs fractional regression for further exploration. Table 8 presents the results of fractional regression. As the TFP level increases, afforestation subsidies, nurturing subsidies, and ecological benefit compensation have a greater positive impact on TFP. However, ecological benefit compensation is not significant at any TFP level. Additionally, as the TFP level increases, the negative impact of forestry regulatory policies on TFP becomes more pronounced. Nevertheless, regardless of the TFP level, the influence of forestry regulatory policies on TFP is not significant. This indicates that forestry subsidies primarily affect high-level TFP, while the impact of forestry regulatory policies on TFP is not significant.

**Table 8.** Fractional regression results.

| | TFP | | |
|---|---|---|---|
| | **Q25%** | **Q50%** | **Q75%** |
| Afforestation subsidies | 0.00758 | 0.0632 *** | 0.130 *** |
| | (0.35) | (2.75) | (2.81) |
| Nurturing subsidies | 0.0325 | 0.0608 *** | 0.103 ** |
| | (1.53) | (2.73) | (2.35) |
| Ecological benefit compensation | −0.00625 | 0.0163 | 0.0727 |
| | (−0.39) | (0.53) | (1.51) |
| Forestry regulatory policies | 0.00119 | −0.0131 | −0.0246 |
| | (0.22) | (−1.42) | (−0.94) |
| Control variable | Controlled | Controlled | Controlled |

Note: t statistics in parentheses, * $p < 0.1$, ** $p < 0.05$, *** $p < 0.01$.

## 6. Robustness Tests

In this paper, we conduct robustness tests by substituting variables and sub-sample regressions, and the estimation results are shown in Table 9. Firstly, the dependent variable was replaced by the total factor productivity measured by the DEA model, including environmental factors, etc. Please refer to Table 9 for the measurement results, and the core explanatory variables such as afforestation subsidies, nurturing subsidies, ecological benefit compensation, and forestry regulatory policies were consistent with the results mentioned earlier. Secondly, 80% of the plot subsamples were randomly selected for the robustness test; please refer to Table 9 model (2). The core explanatory variables are largely consistent with the results mentioned earlier. Finally, the plots were divided into two sub-samples of plots, public welfare forests, and commercial forests, based on the classified forestry management system, to test the robustness of the effects of forestry subsidies and forestry regulatory policies on the total factor productivity of the two sub-samples. According to model (3) in Table 9, the research findings are largely consistent with those mentioned earlier. According to model (4) in Table 9, afforestation subsidies and conservation subsidies are significantly and positively related to the total factor productivity of commercial forests. Since commercial forest plots are not the target of ecological benefit compensation, this variable was automatically deleted from model (4) in Table 9. The research findings are largely consistent with those mentioned earlier. Robustness tests indicate that the research results of this paper are fundamentally stable.

**Table 9.** Robustness test results.

| | **(1) Total Sample** | **(2) Random Sample Sub-Samples** | **(3) Public Forest Sub-Samples** | **(4) Commercial Forest Sub-Samples** |
|---|---|---|---|---|
| Afforestation subsidies | 0.0751 *** | 0.0715 ** | 0.169 ** | 0.0565 *** |
| | (2.81) | (2.56) | (2.28) | (2.62) |
| Nurturing subsidies | 0.0666 ** | 0.0982 *** | −0.0785 | 0.0710 *** |
| | (2.41) | (3.30) | (−1.02) | (3.20) |
| Ecological benefit compensation | 0.0242 | 0.00537 | 0.0504 | 0 |
| | (0.61) | (0.12) | (1.16) | (.) |
| forestry regulatory policies | −0.00963 | −0.0269 | 0.00790 | 0.00592 |
| | (−0.47) | (−1.22) | (0.21) | (0.36) |
| Control variables | Controlled | Controlled | Controlled | Controlled |
| LR chi2(10) | 50.53 | 42.08 | 15.10 | 37.17 |
| Prob > chi2 | 0.000 | 0.000 | 0.088 | 0.000 |
| N | 295 | 236 | 31 | 264 |

Note: t statistics in parentheses, * $p < 0.1$, ** $p < 0.05$, *** $p < 0.01$.

## 7. Conclusions

### 7.1. Findings of the Study

Based on the full-cycle survey data of 295 forest plots in Liuyang City, Hunan Province, this paper measures the plot-scale total factor productivity of forestry excluding the influence of environmental factors, and explores the impact and differences of forestry subsidies and forestry regulatory policies on the total factor productivity of timber forests, economic forests, and mixed forests. The results of the study show the following: (1) There are differences in total factor productivity in forestry at different measurement scales, and the mean value of total factor productivity in forestry at the plot scale is 0.127, which is lower than the efficiency of forestry production at the provincial and household scales as measured in the existing literature. There are apparent differences in total factor productivity of heterogeneous forest types, with the following relationship: timber forest > economic forest > mixed forest. (2) Afforestation subsidies and nurturing subsidies significantly positively influence high-level TFP; and ecological benefit compensation positively affects high-level TFP, but is not significant at any level of TFP. Forestry regulatory policies negatively impacts high-level TFP, but are not significant at any level of TFP. (3) The comparison of empirical estimation results for heterogeneous forest types shows that afforestation subsidies have a significant positive effect on the total factor productivity of all three forest types. Nurturing subsidies have a significant positive effect on the total factor productivity of economic forests only. Ecological benefit compensation, on the other hand, does not have a significant effect on the total factor productivity of any of the three forest types, but it is negatively correlated with the total factor productivity of timber forests and economic forests. Forestry regulatory policies had no significant effect on the total factor productivity of all three forest types but were negatively related to the total factor productivity of timber forests.

### 7.2. Policy Recommendations

According to the results of this paper, the total factor productivity of various types of forest types in collective forest areas has ample space for improvement. The potential for the excavation of forest products in collective forest areas is still enormous and should be improved and adjusted based on the accurate anchoring of the policy role of the object of the forestry subsidy and forestry regulatory policies. This will improve the total factor productivity of various types of forest types and enhance the supply capacity of forestry in collective forest areas.

Concerning forestry subsidies, firstly, the universality of afforestation subsidies should be expanded. Afforestation costs are an essential input cost for all types of forests, and afforestation subsidies are a regular policy tool in forestry in developed countries to support forestry development, so afforestation subsidy funds can reach more small-scale farmers, thereby enhancing their incentives to engage in forestry. The paper's empirical results confirm that afforestation subsidies have a significant positive impact on the total factor productivity of timber forests, economic forests, and mixed forests. However, the descriptive analysis shows that only 17.63 percent of the sample plots received afforestation subsidies. Therefore, expanding the universality of afforestation subsidies for timber forests, economic forests, mixed forests, and other types of forests will help provide factor productivity in forestry. Secondly, we should improve the precision of the nurturing subsidies. According to the results of the empirical research in this paper, the nurturing subsidy has a significant positive effect on the total factor productivity of economic forests. It has a positive effect on the total factor productivity of timber and mixed forests, but it is insignificant. According to the Forest Nurturing Regulations, economic forests require nurturing management at the young forest stage, early fruiting stage, and complete fruiting stage. However, nurturing at the young forest stage and early fruiting stage requires more significant inputs and produces less output. Thirdly, the scale of public welfare forests and commercial forests in collective forest areas should be planned in an integrated manner. According to the empirical findings of this paper, the ecological compensation benefits have a negative impact on the total factor productivity of timber forests and economic forests. The Programme

for Deepening the Reform of the Collective Forest Rights System states that the scope of public welfare forests and natural forests should be scientifically delineated. The scope should not be expanded arbitrarily. By reasonably optimizing the proportion of collective forests in public welfare forests, appropriate consideration should be given to transferring collective forest land with an unimportant forest ecological location or a non-vulnerable ecological situation out of the scope of public welfare forests by the law. Therefore, the scale of public welfare forests and commercial forests should be planned in an integrated manner to promote the scientific and rational management and use of all types of forests in collective forest areas on the premise of safeguarding ecological security, improving total factor productivity in forestry, promoting rural revitalization, and, in turn, raising the incomes of farming households.

Regarding the control of forestry production, after the new round of reform of the collective forest tenure reform, the Government has continued to deepen the reform of "decentralization, simplification of management procedures, and optimization of services" in the control of forest logging, but the results of empirical research still show that the forestry regulatory policies negatively affects the total factor productivity of timber forests. It should be under the premise that safeguarding ecological safety, promoting the transformation of the government's function from "forest logging regulatory" to "forest logging service", safeguarding the right of forest operators to dispose of forest trees, and promoting the better realization of the market value of timber and other forest products will help to improve the total factor productivity of collective forest areas. This will help to increase the total factor productivity of collective forest areas.

**Author Contributions:** Conceptualization and writing—original draft, L.C.; methodology and software, C.J.; formal analysis and investigation, Q.X.; revise, T.X., S.L., J.H. and S.P. All authors have read and agreed to the published version of the manuscript.

**Funding:** 1. Changsha Municipal Science and Technology Bureau Soft Science Key Project: Research on Government-led Forest Right Storage Mode and Realization Path in the Context of "Double Carbon" (KH2302041); and 2. Key Issues of Hunan Social Science Review Committee: Research on the Utilization Efficiency, Influencing Factors and Policy Optimization of Collective Forest Land in the Context of Rural Revitalization (XSP22ZDI022).

**Data Availability Statement:** Our data is confidential government data and is not readily available for publication.

**Acknowledgments:** We thank all external reviewers and editors. All the article's problems are the authors' responsibility.

**Conflicts of Interest:** The authors declare no conflict of interest.

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
