# Peer review of "Forestry Subsidies, Forestry Regulatory Policies, and Total Factor Productivity in Forestry—Plot-Scale Micro-Survey Data from A Heterogeneous Forest Types Perspective"

_forests, doi:10.3390/f15040692_

Round 1

Reviewer 1 Report

Comments and Suggestions for Authors

I think this article is a valuable study in terms of optimizing forestry production control policy, and improve forestry total factor productivity from the perspective of heterogeneous forest species.

Author Response

Dear Reviewer,

We would like to express our sincere gratitude for your thorough review and positive evaluation of our manuscript titled "Forestry Subsidies, Forestry Regulatory Policies and Total Factor Productivity in Forestry——Plot-scale Micro-survey Data from A Heterogeneous Forest Types Perspective." Your favorable comments and high assessment of our work are deeply appreciated.

Your recognition of the quality and significance of our research is truly motivating for us. Your feedback reinforces our confidence in the value of our study and its contribution to the field. We are grateful for your recognition of the efforts we have invested in this work.

We also want to thank you for taking the time to assess our manuscript comprehensively. Your insights and expertise have undoubtedly enriched the quality of our research. Your positive feedback encourages us to continue striving for excellence in our academic endeavors.

Best regards.

Reviewer 2 Report

Comments and Suggestions for Authors

Dear authors of this manuscript.

I do not feel I have the best background to evaluate this manuscript. I am an ecologist with good knowledge on foretsry practies in general from our part of the world - but I am not familiar with economic "language and methods".

With this background it is difficult to have any strong opinion on the quality of the manuscript as such.

But I can recommend some changes; - first if all -this is written by and for chinese foresters/econmists  to make advices for improvement in forestry practices by politicians. 

That makes is more difficult to catch for us outside China - mainly because we know so litle about this politics and culture in your country.

I would at least like you to explain this better in the Intruduction part of the text:

Some terms to be explained for us that knows litle about Chinese forests and forestry is:

"Collective forest area"

 How and who does "Productivity control" ?

The concept "Total factor productivity " ( How is it evalutated/calcultated ?)

You operate with three catecgories of forests: Explain it in the Introduction/Methods - it comes somehow in a table.

"Heterogenous forest species" is a kind of forest management area - but it includes then all biodiversity - does you mean a kind of mixed tree species forest ? How does it look alike ?

I always recommend to put in photoes to explaini a litle the kind of habitats/forests that is studied - can it be done as examples ?

Is there  any concern at all about biodiversity protecticon philosophy/preactise involved here in Chinese forestry ?

If so - it could be mentioned - also if this is not an aspect at all -

But the conctrol for forestry practise in different kinds of certification systems is common in Europe and North America - and could be mentioned in a way ?

I have noticed - but not controlled - the references. They are all/mainly written  by Chines authors - OK as this is meant for China -  I also positively notes many referred papers are in International Journals -

Comments on the Quality of English Language

I think the English is OK - but it is many long sentences and as such hard to understand properly without reading it twice or more. This could be improved - just make shorter sentences and evaluate what is essential to write. 

Reviewer 3 Report

Comments and Suggestions for Authors

The article is focusedon an interesting topic. Subsidies have always been a problem in forest policy. Overall, the manuscript is well prepared but needs improvement in some parts to highlight the authors' contribution in their article. 

1. The text mentions concepts such as timber forests, economic forests, and mixed forests. It is clear that forests provide both economic and ecosystem services. Please clarify the nature of these forests before Table 3 (lines 150-151).

2. What is the difference in pine plantations between timber forests and mixed forests?

3. The study simultaneously used DEA, Stochastic Frontier, and Tobit regression. Why dod you use simultaneously  DEA and Stochastic Frontier? In the literature, they are mentioned as alternative methods. Also, what is the production function used in Stochastic Frontier Analysis? The literature on this topic state that the advantages of DEA are in its non-parametric nature.

4. What necessitates the use of Tobit regression? Tobit has long been considered an unacceptable method in specialized literature. Fractional regression is recommended instead. Was a bootstrap procedure conducted with Tobit?

5. The citation is not provided for the total factor productivity formula (lines 151-155). When using DEA, there is a well-established methodology for this, namely the Malmquist index. Usually, when using DEA without studying changes over time in the influence of factors on the result, it is more appropriate to use the term efficiency because that is what DEA actually measures.

6. The literature used is quite limited for a study of this magnitude. There should be significant development in the literature review regarding the methodology of the study and research related to productivity, efficiency, and the factors influencing them in forestry.

7. Please review the tables. They content chinese symbols.

All the mentioned improvement directions require a bit more explanation regarding the reasons for their use in the text.

Wish the authors great success.

Round 2

Reviewer 3 Report

Comments and Suggestions for Authors

Dear authors,

Thank you for your kind reply and profound explanations of your research. You have addressed the necessary remarks in the improved manuscript.

Thank you for that.